# Programmable human histone phosphorylation and gene activation using a CRISPR/Cas9-based chromatin kinase

Jing Li[1], Barun Mahata[1], Mario Escobar [2], Jacob Goell[1], Kaiyuan Wang [1], Pranav Khemka[1] & Isaac B. Hilton [1,2 ✉]

Histone phosphorylation is a ubiquitous post-translational modification that allows eukaryotic cells to rapidly respond to environmental stimuli. Despite correlative evidence linking histone phosphorylation to changes in gene expression, establishing the causal role of this key epigenomic modification at diverse loci within native chromatin has been hampered by a lack of technologies enabling robust, locus-specific deposition of endogenous histone phosphorylation. To address this technological gap, here we build a programmable chromatin kinase, called dCas9-dMSK1, by directly fusing nuclease-null CRISPR/Cas9 to a hyperactive, truncated variant of the human MSK1 histone kinase. Targeting dCas9-dMSK1 to human promoters results in increased target histone phosphorylation and gene activation and demonstrates that hyperphosphorylation of histone H3 serine 28 (H3S28ph) in particular plays a causal role in the transactivation of human promoters. In addition, we uncover mediators of resistance to the BRAF V600E inhibitor PLX-4720 in human melanoma cells using genome-scale screening with dCas9-dMSK1. Collectively, our findings enable a facile way to reshape human chromatin using CRISPR/Cas9-based epigenome editing and further define the causal link between histone phosphorylation and human gene activation.

[1] Department of Bioengineering, Rice University, Houston, TX, USA. [2] Department of BioSciences, Rice University, Houston, TX, USA. ✉email: isaac.hilton@rice.edu

D ynamic epigenomic regulatory forces, including DNA methylation and post-translational modifications (PTMs) to histones, harmonize to control human gene expression[1–3]. Histone phosphorylation at serine residues 10 and 28 on histone subunit H3 (H3S10ph and H3S28ph, respectively), is one type of histone PTM that has been correlated with stimulus-dependent gene expression[4–9]. Despite this correlation, defining the causal function of endogenous H3S10ph and H3S28ph has been challenging due to a lack of technologies to manipulate histone phosphorylation at diverse loci within native chromatin contexts.

The mitogen- and stress-activated protein kinase 1 (MSK1) is one of nine human proteins known to catalyze H3S10ph and H3S28ph in vitro[10]. MSK1 is primarily localized to the nucleus, where it can be activated by ERK or p38 mitogen-activated protein kinase (MAPK)-mediated phosphorylation and subsequent autophosphorylation[11,12]. MSK1-driven H3S10ph/H3S28ph has been correlated with the transactivation of stimulus-responsive genes through chromatin immunoprecipitation (ChIP) assays[4,5]. Furthermore, genome-wide analyses of H3S28ph levels suggest that MSK1-mediated histone phosphorylation is tightly linked to gene expression from most stress-responsive human promoters[8]. In addition, the artificial recruitment of MSK1 to endogenous NF1 transcription factor binding sites using an NF1-MSK1 fusion protein results in hyperphosphorylation of nearby histone H3S10 and H3S28 residues and increased gene expression from adjacent promoters[13].

The CRISPR/Cas9 system has been repurposed for programmable genome editing in human cells[14–16]. In parallel, nuclease-null deactivated CRISPR/Cas platforms have also been developed to manipulate endogenous histone PTMs at targeted loci[17–22]. However, no CRISPR/Cas-based tools have been created that permit the locus-specific modification of histone phosphorylation.

Here we construct a fusion protein, called dCas9-dMSK1, that consists of the deactivated Cas9 protein from *Streptococcus pyogenes* (dCas9) and a hyperactive variant of human MSK1. We show that dCas9-dMSK1 permits locus-specific manipulation of H3S10ph and H3S28ph at targeted human loci and in turn the activation of gene expression from human promoters. Our work demonstrates that histone phosphorylation plays a causal role in the activation of human promoters, establishes a new way to engineer the human epigenome, and expands the dCas9-based epigenome editing arsenal.

## Results

**Development of a CRISPR/Cas9-based histone kinase.** Human MSK1 has been linked to the deposition of H3S10ph/H3S28ph and transcriptional activation[7,9–13], and our preliminary results showed that knockout of MSK1 causes changes to RNA polymerase II-mediated transcription in human cells (Supplementary Fig. 1, Supplementary Tables 1 and 2). Therefore, we hypothesized that dCas9 could be used to recruit MSK1 to individual human loci and clarify the casual role that locus-specific histone phosphorylation plays in human gene expression. To test this hypothesis, we constructed three fusion proteins between the C-terminus of dCas9 and the N-terminus of different MSK1 variants (Fig. 1a). Specifically, we fused dCas9 to full-length WT MSK1 (dCas9-MSK1), to MSK1 lacking an N-terminal inhibitory domain (NID; Supplementary Fig. 2) that has previously been observed to limit catalytic activity on chromatin assembled in vitro (dCas9-dMSK1), and to a catalytically inactivated version of dMSK1 (dCas9-ddMSK1)[10]. dCas9-MSK1 and dCas9-dMSK1 both displayed hallmarks of natural MSK1 catalytic activity (autophosphorylation of MSK1 serine residues 212 and 376)[12] when transfected into human cells, whereas the catalytically

inactive dCas9-ddMSK1 fusion did not (Fig. 1b). Moreover, both dCas9-MSK1 and dCas9-dMSK1 were able catalyze H3S10ph and H3S28ph when incubated with in vitro assembled human histone octamers (Fig. 1c), however, dCas9-dMSK1 exhibited a significantly ($P = 0.029$) greater level of enzymatic activity on H3S28 than dCas9-MSK1 (Fig. 1d). Together these results demonstrate that dCas9-MSK1 and dCas9-dMSK1 are catalytically active in human cells, able to phosphorylate human histones in vitro in the absence of auxiliary cellular cofactors, and that dCas9-dMSK1 harbors hyperactive histone H3S28 phosphorylation kinase activity relative to dCas9-MSK1.

**dCas9-dMSK1 activates natural MSK1 targets.** To test whether the catalytically active dCas9-MSK1 and dCas9-dMSK1 fusion proteins could modulate endogenous human histone phosphorylation and/or gene expression, we first identified the natural targets of MSK1 using comparative RNA-seq. Two independent MSK1 knockout (KO) clones were generated by disrupting conserved exons in MSK1 (exons 1 and 2, respectively) using Cas9-mediated KO (Supplementary Fig. 1a,b). Each clonal MSK1 KO cell line displayed strikingly similar transcriptome-wide changes compared to WT HEK293T cells (Supplementary Fig. 3). Together, comparative RNA-seq between WT and MSK1 knockout cell lines resulted in 41 and 24 shared (between both knockout lines) downregulated and upregulated genes, respectively (Fig. 2a).

We next designed guide RNAs (gRNAs) to recruit each fusion protein to the promoter regions of the top five most significantly (adjusted $P$ value <0.01) downregulated genes (*PRKCB*, *BMP2*, *SHROOM2*, *ZNF462*, and *GDF6*) in MSK1 KO HEK293T cells (Supplementary Fig. 4) to test whether dCas9-MSK1 and/or dCas9-dMSK1 could activate natural MSK1 targets (i.e., genes downregulated upon the loss of MSK1). RT-qPCR revealed that dCas9-dMSK1, but not dCas9-MSK1, significantly activated *BMP2* ($P = 0.002$) and *GDF6* ($P = 0.008$) relative to a dCas9 control when targeted to each respective promoter region using specific gRNAs (Fig. 2b). Notably, dCas9-MSK1 was unable to activate any target genes, which we attribute to MSK1's reliance upon co-activators (e.g., ATF and/or CREB) to penetrate endogenous human chromatin and phosphorylate histones, a requirement that dMSK1 (lacking the NID) does not appear to share[10]. This capability of dCas9-dMSK1 could be instrumental in the analysis of regulatory mechanisms controlling stimulus-induced transcription, for instance, by studying the role of CREB/ATF1 phosphorylation and recruitment of associated reader proteins in response to MSK1 targeting. Not all genes that were downregulated upon the loss of MSK1 were responsive to dCas9-dMSK1-mediated gene activation (Fig. 2b) suggesting that some downregulated genes were either indirect targets of MSK1 (i.e., regulated instead by direct targets of MSK1) and/or were not influenced by dCas9-dMSK1 mediated epigenetic changes at the gRNA sites targeted. To measure MSK1 occupancy at *PRKCB*, *BMP2*, *SHROOM2*, *GDF6*, and *ZNF462*, we performed ChIP-qPCR at the promoter region (Supplementary Fig. 4) of each gene in WT and MSK1 KO HEK293T cells. These assays demonstrate that MSK1 is significantly enriched at the promoter regions of *BMP2* ($P = 0.0001$) and *GDF6* ($P = 0.0009$), but not at the promoter regions of *PRKCB*, *SHROOM2*, or *ZNF462* (Supplementary Fig. 5). Coupled with the fact that *BMP2* and *GDF6* can be activated by dCas9-dMSK1, whereas *PRKCB*, *SHROOM2*, and *ZNF462* cannot, these results support the hypothesis that *BMP2* and *GDF6* are direct targets of MSK1, whereas *PRKCB*, *SHROOM2*, and *ZNF462* appear to be indirectly affected by the loss of MSK1.

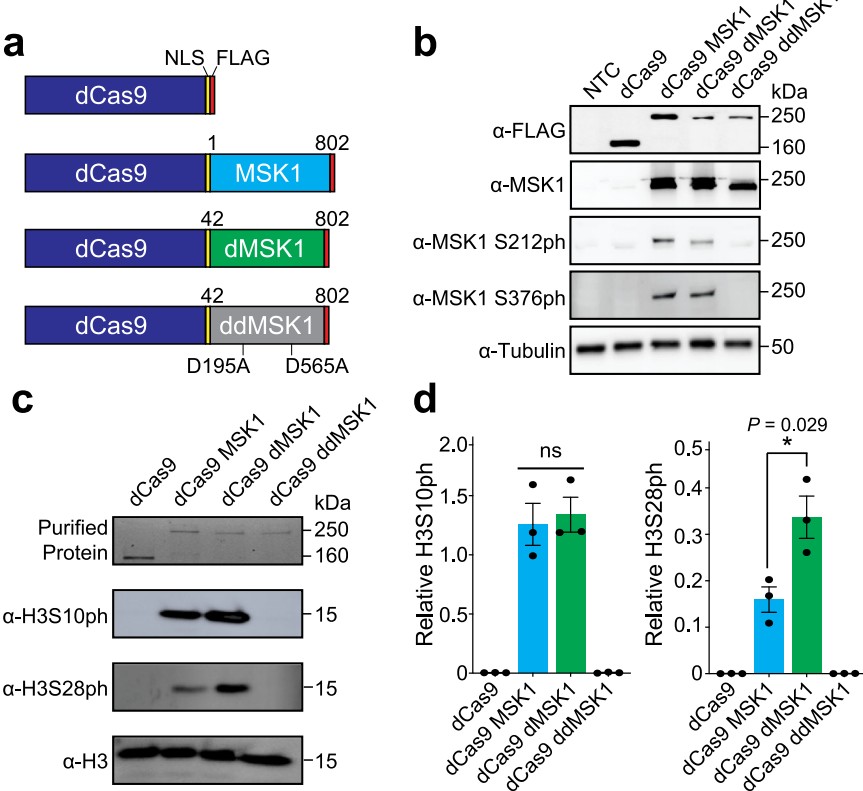

**Fig. 1 Development of a CRISPR/Cas9-based histone kinase. a** Schematics of dCas9, dCas9 fused to WT MSK1 (dCas9-MSK1), dCas9 fused to a hyperactive truncated MSK1 (dCas9-dMSK1), and dCas9 fused to a catalytically inactivated version of dMSK1 (dCas9-ddMSK1). **b** dCas9 and dCas9-MSK1 fusion variants were transiently transfected into HEK293T cells and autophosphorylation levels were detected (at serine residues 212 and 376 of MSK1) by Western blot 72 h post-transfection. Data are representative of three independent biological experiments. **c** Purified dCas9, dCas9-MSK1, dCas9-dMSK1, and dCas9-ddMSK1 were incubated with human histone octamers in vitro and resulting levels of histone H3 phosphorylation at serine residues 10 and 28 (H3S10ph and H3S28ph, respectively) were measured after incubation for 1 h. **d** The relative levels of H3S10ph and H3S28ph were quantified using densitometry from three different in vitro histone kinase assays. Two-sided t-test, *$P < 0.05$; $n = 3$ independent experiments in panel (**d**); error bars, s.e.m.; ns, not significant; kDa, kilodaltons. Source data are available in the Source data file.

To further investigate the mechanisms underlying dCas9-dMSK-mediated gene activation at responsive (*BMP2*) and non-responsive (*PRKCB*) loci, we used ChIP-qPCR to measure histone phosphorylation levels. Interestingly, despite high levels of dCas9-dMSK1 binding (Supplementary Fig. 6), the *PRKCB* promoter displayed no changes in H3S10ph nor H3S28ph in response to dCas9-dMSK1 targeting (Fig. 2c). In contrast, dCas9-MSK1 and dCas9-dMSK1 both significantly ($P = 0.012$ and $P = 0.008$, respectively) elevated H3S10ph levels at the *BMP2* promoter compared to targeting with a dCas9 control (Fig. 2d). However, only dCas9-dMSK1 efficiently phosphorylated both H3S10 and H3S28 at the *BMP2* promoter (Fig. 2d). These results at endogenous histones mirror the findings using histone octamers in vitro (Fig. 1d) and demonstrate that dCas9-dMSK1 is more efficient than dCas9-MSK1 at phosphorylating histone H3S28 both in vitro and within native human chromatin. Furthermore, these results indicate that targeted, locus-specific H3S28ph is causal for the activation of gene expression from human promoters that are sensitive to histone phosphorylation.

**dCas9-dMSK1 activates human promoters with high specificity.** To determine the effects of dCas9-dMSK1 on an expanded set of target genes and genomic regulatory elements, we delivered dCas9-fusion proteins to the distal enhancer (DE), proximal enhancer (PE), and proximal promoter (PP) of *OCT4* (Supplementary Fig. 7a). RT-qPCR 72 h after transient transfection in

HEK293T cells showed that dCas9-dMSK1 was capable of potently activating *OCT4* expression (Fig. 3a), but that this potency was concentrated to ~260–103 bp upstream of the *OCT4* transcription start site (TSS; Supplementary Fig. 7b). We also targeted dCas9 and dCas9-MSK1 fusion proteins to the distal regulatory region (DRR), core enhancer (CE), and promoter of *MYOD* (Supplementary Fig. 8a) and observed a similar potency of dCas9-dMSK1 driven gene activation that was localized to the *MYOD* promoter region (~273–35 bp upstream of the *MYOD* TSS; Fig. 3b, Supplementary Fig. 8b). These data show that dCas9-dMSK1 can synthetically activate non-natural MSK1 target genes and that human promoters appear to be more sensitive to the regulatory effects of endogenous histone phosphorylation than distal enhancers.

To assess the specificity of dCas9-dMSK1, we performed ChIP-seq and RNA-seq in HEK293T cells co-transfected with dCas9-dMSK1 and four *OCT4*-targeting gRNAs or with dCas9-dMSK1 and a non-targeting gRNA control used previously[23]. Our ChIP-seq results showed that dCas9-dMSK1 binding to the *OCT4* promoter was highly specific (FDR = 0.0006) across the human genome (Fig. 3c), with only one significant (FDR = 0.0086) off-target within an intron of *CROCC2* identified. RNA-seq demonstrated that dCas9-dMSK1 targeted to the *OCT4* promoter specifically and significantly activated two different isoforms of *OCT4* (ENST00000638788.1 and ENST00000461401.1; $P_{adj}$ of 0.046 and 0.008, respectively) across the human transcriptome (Fig. 3d). One off-target transcript was detected upon dCas9-

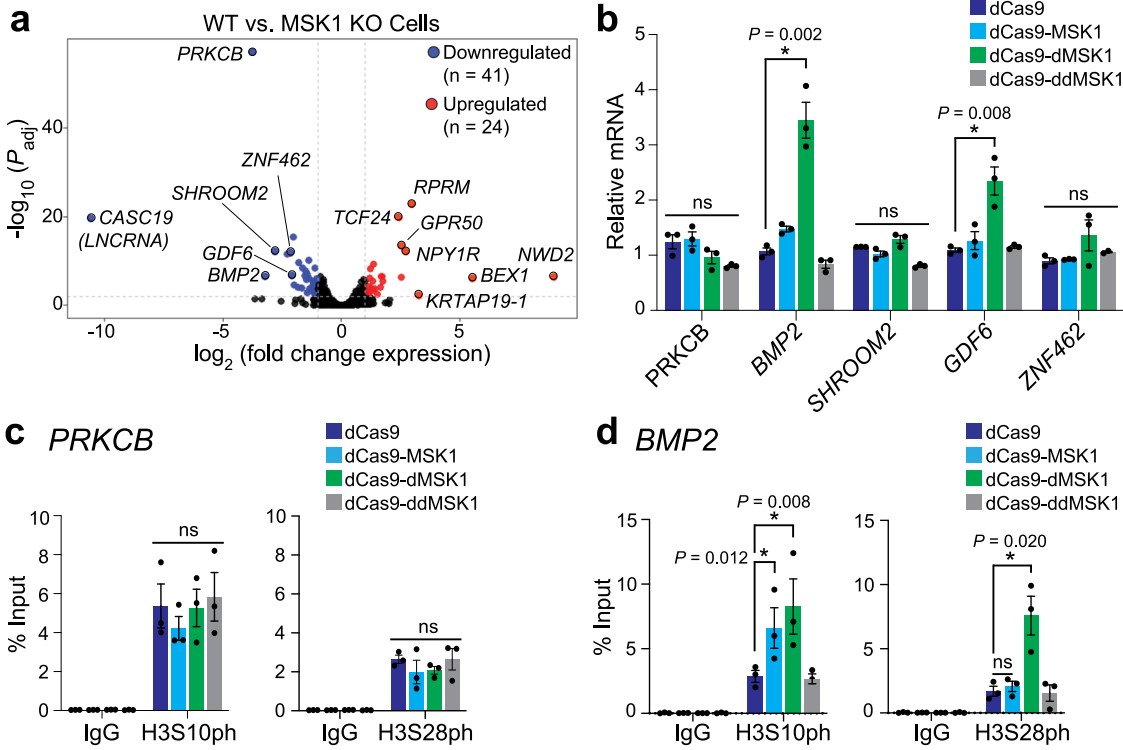

**Fig. 2 dCas9-dMSK1 activates natural MSK1 target genes. a** Differentially expressed genes in MSK1 knockout HEK293T cell lines compared to WT HEK293T cells are shown (blue circles, downregulated genes; red circles, upregulated genes). Data were analyzed using the Wald test and the adjusted $P$ value ($P_{adj}$) was calculated using the Benjamini and Hochberg method. **b** mRNA levels for the top five downregulated genes were measured by RT-qPCR at 72 h post-transfection of indicated dCas9-fusion proteins (or dCas9 control) and corresponding gRNAs. **c**, **d** ChIP-qPCR for H3S10ph and H3S28ph at the PRKCB and BMP2 promoters, respectively, 72 h post-transfection with the indicated dCas9-fusion proteins or dCas9 control. Two-sided $t$-test, *$P <$ 0.05; $n = 3$ independent experiments for panels (**b–d**); error bars; s.e.m.; ns, not significant. Source data are available in the Source data file.

dMSK1 targeting to the OCT4 promoter in HEK293T cells; SEPT7P3 (Septin 7 pseudogene 3), which was significantly ($P_{adj} = 0.011$) downregulated. Nevertheless, our results collectively demonstrate that dCas9-dMSK1 binding and activation of promoters are highly specific across the human genome and transcriptome, respectively.

Consistent with our observations above, dCas9-MSK1 and dCas9-dMSK1 both significantly ($P$ value <0.05) increased H3S10ph levels compared to dCas9 when targeted to the OCT4 and MYOD promoters (Fig. 4a, b). However, only dCas9-dMSK1 resulted in increased H3S10ph and H3S28ph levels at these targeted promoters, which was coincident with the activation of gene expression (Fig. 3a, b). H3S28ph ChIP-seq analysis confirmed our ChIP-qPCR observations that H3S28ph was enriched at the OCT4 promoter subsequent to dCas9-dMSK1 targeting, however, this H3S28ph enrichment was not statistically significant above background on a genome-wide scale (using cutoffs of log2 fold change >1 and FDR < 0.01; Supplementary Fig. 9). Interestingly, previous studies suggest that MSK1 phosphorylates either H3S10 or H3S28 but not both upon the same histone tail[24–26]. To evaluate if dCas9-dMSK1 could artificially deposit both H3S10ph and H3S28ph on the same histone tail, we performed re-ChIP analysis (Fig. 4c, d). In agreement with previous findings, our re-ChIP analyses indicated that dCas9-dMSK1-mediated phosphorylation of H3S10 occurred independently of H3S28 phosphorylation at the OCT4 and MYOD promoters (Fig. 4c, d, respectively), as evidenced by the lack of enrichment of H3S28ph after specific enrichment of H3S10ph via ChIP. Collectively our results show that H3S28ph in particular plays a causal role in the activation of gene expression from natural and non-natural MSK1 target genes, and moreover,

highlights the specificity and mechanistic utility of dCas9-dMSK1.

**dCas9-dMSK1 driven H3S28ph influences local H3K27ac status.** H3S28ph has previously been shown to influence, and be influenced by, the dynamics of surrounding histone PTMs, especially the acetylation status of histone H3 lysine 27 (H3K27ac)[4,8,9,13,26]. Therefore, we also measured H3K27ac levels after targeting the OCT4 and MYOD promoters with dCas9 and dCas9-MSK1 fusion protein variants. Despite having no intrinsic histone acetyltransferase activity, targeting dCas9-dMSK1 to the OCT4 and MYOD promoters resulted in increased H3K27ac levels at both loci (Fig. 5a, b). Similarly, H3K27ac was enriched when dCas9-dMSK1 was targeted to the BMP2 promoter, but not when targeted to the PRKCB promoter (Supplementary Fig. 10). These results show that crosstalk exists between H3S28ph and H3K27ac at endogenous human loci and suggest that the interplay between H3S28ph and H3K27ac can result in the activation of gene expression from human promoters. Moreover, these observations are consistent with reports showing that H3S28ph promotes CBP/p300-dependent transcription in cell-free systems[9].

To further investigate the relationship between histone phosphorylation and other histone PTMs, we measured how chemical inhibition of CBP/p300 histone acetyltransferases impacted dCas9-dMSK1 driven gene activation. Specific inhibition of CBP/p300 activity using A485 significantly reduced the efficacy of dCas9-dMSK1-mediated transactivation of OCT4 ($P =$ 0.008) and MYOD ($P =$ 0.0008) promoters (Fig. 5c, d), demonstrating that the activity of CBP/p300 appears to be functionally

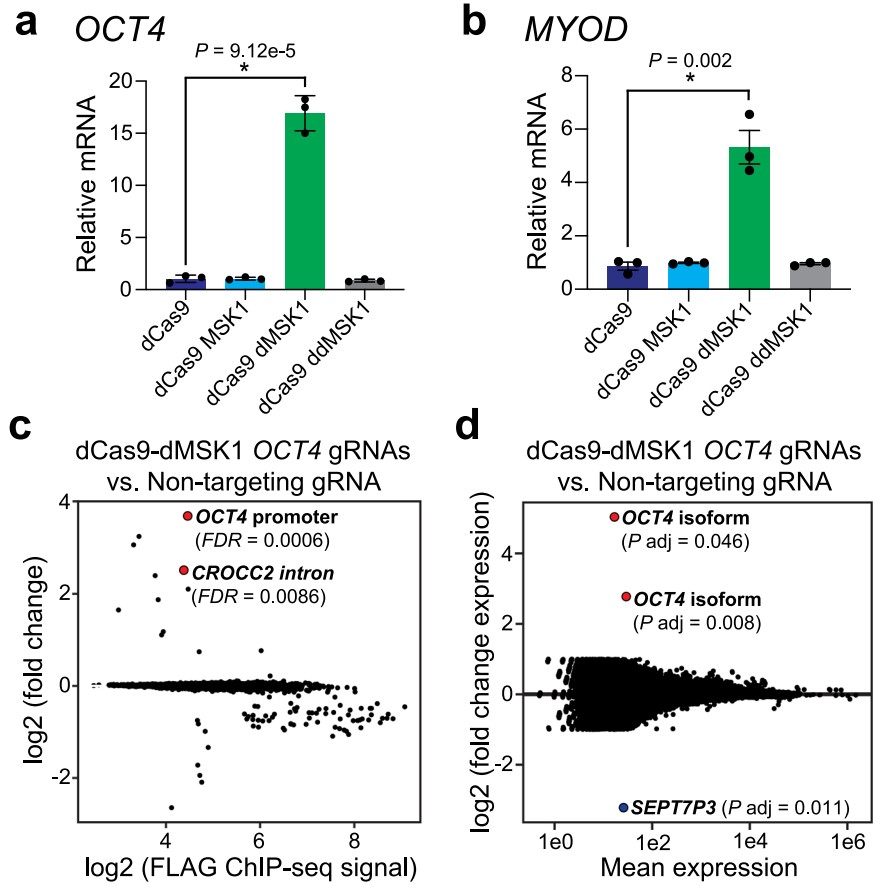

**Fig. 3 dCas9-dMSK1 activates endogenous human promoters with high genome and transcriptome-wide specificity. a, b** RT-qPCR for *OCT4* or *MYOD* mRNA levels 72 h post-transfection of dCas9, dCas9-MSK1, dCas9-dMSK1, or dCas9-ddMSK1 and 4 corresponding promoter-targeting gRNAs. Two-sided *t*-test, *$P < 0.05$; $n = 3$ independent experiments for both panels; error bars, s.e.m. **c** DESeq2 analysis of FLAG ChIP-seq binding data from HEK293T cells transiently co-transfected with dCas9-dMSK1 and four *OCT4* promoter-targeting gRNAs compared to HEK293T cells transiently co-transfected with dCas9-dMSK1 and a non-targeting gRNA. Data were analyzed using the Wald test. Red circles indicate false discovery rate (FDR) < 0.01; **d** DESeq2 analysis of RNA-seq data from HEK293T cells transiently co-transfected with dCas9-dMSK1 and four *OCT4* promoter-targeting gRNAs compared to HEK293T cells transiently co-transfected with dCas9-dMSK1 and a non-targeting gRNA. Data were analyzed using the Wald test and the adjusted *P* value ($P_{adj}$) was calculated using the Benjamini and Hochberg method. Significantly upregulated mRNAs (*OCT4* isoforms) are shown as red circles ($P_{adj} < 0.05$) and a significantly downregulated gene (*SEPT7P3*) is designated with a blue circle ($P_{adj} < 0.05$). Source data are available in the Source data file.

linked to the transactivation capacity of dCas9-dMSK1 at endogenous human promoters. In addition, WT MSK1 has been shown to physically and functionally interact with the KMT2A/MLL1 methyltransferase complex, which can catalyze methylation of H3K4 (ref. [27]). Therefore, we measured the enrichment of H3K4me3 after targeting the *OCT4* and *MYOD* promoters with dCas9 or dCas9-MSK1 fusion proteins (Supplementary Fig. 11). Although we did not observe any significant ($P < 0.05$) increases in H3K4me3 enrichment after targeting these promoters with dCas9-MSK1 nor dCas9-dMSK1, we note that as synthetic, programmable dCas9-based histone kinases, dCas9-MSK1 and dCas9-dMSK1 may not mechanistically function in exactly the same way(s) as WT MSK1 in human cells.

We also targeted dCas9-fusion proteins to the *HBA1, SOX2,* and *KLF4* promoter regions (Supplementary Fig. 12a–c) in HEK293T cells and measured gene expression 72 h post transient transfection (Supplementary Fig. 12d–f). dCas9-dMSK1 significantly ($P$ value <0.05) activated gene expression from each targeted promoter, although to a lesser extent than *OCT4* or *MYOD*. The reduced potency at these loci may be due in part, to differences in basal gene expression levels (Supplementary Table 3), as has been observed previously with dCas9-based activators[28]. Interestingly, each additional gRNA used to recruit dCas9-dMSK1 to a target promoter resulted in additive increases

in gene activation, suggesting that local levels of histone phosphorylation are directly proportional to gene expression at responsive promoters (Supplementary Fig. 13). We also found that dCas9-dMSK1 was able to activate gene expression in A549 cells (Supplementary Fig. 14), and furthermore, that another human histone kinase (Aurora B) could also activate gene expression when fused to dCas9 and targeted to a human promoter (Supplementary Fig. 15). Altogether, these results show that dCas9-dMSK1 can be used to engineer the phosphorylation status of endogenous human histones which in turn can result in the activation of gene expression from human promoters. This capability is functional at diverse human promoters and in different human cell types, and can be achieved using other histone kinases, further establishing the causal role and functional importance histone phosphorylation at endogenous human loci.

**Genome-scale screening for mediators of PLX-4720 resistance.** dCas9-based tools can be used in unbiased approaches to screen the noncoding genome in high-throughput[28,29]. To test the efficacy of dCas9-dMSK1 at thousands of human promoters in high-throughput and to evaluate if targeted histone phosphorylation could uncover mediators of pathological gene expression, we combined dCas9-dMSK1 with a genome-scale gRNA library to

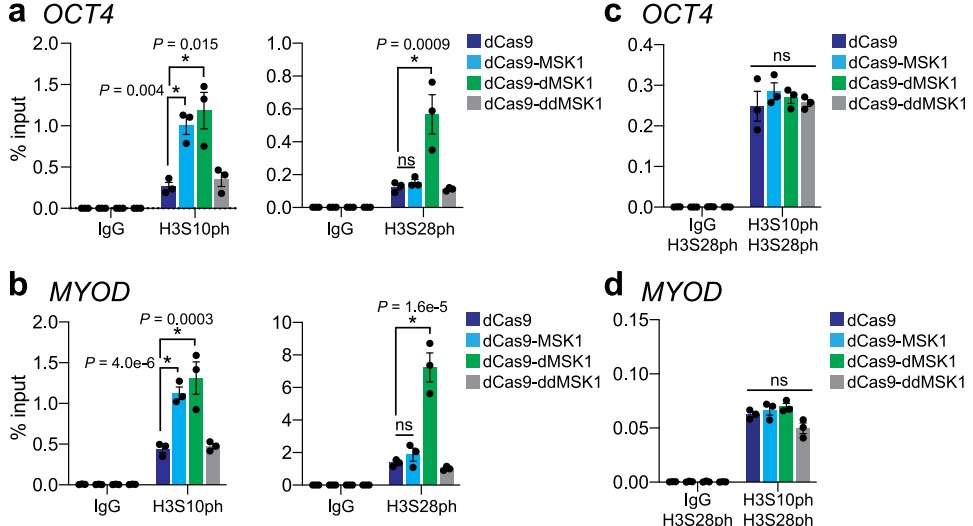

**Fig. 4 dCas9-dMSK1 induces target gene activation via H3S28ph. a, b** ChIP-qPCR enrichment for H3S10ph and H3S28ph, at the *OCT4* (panel **a**) or *MYOD* (panel **b**) promoters 72 h post-transfection of dCas9 or indicated dCas9-fusion protein and corresponding gRNAs. **c, d** Re-ChIP-qPCR enrichment for H3S10ph and H3S28ph at *OCT4* (panel **c**) or *MYOD* (panel **d**) promoters 72 h post-transfection of dCas9 or indicated dCas9-fusion protein and corresponding gRNAs. Two-sided *t*-test, *\*P < 0.05; n = 3 independent experiments for all panels; error bars, s.e.m.; ns, not significant. Source data are available in the Source data file.

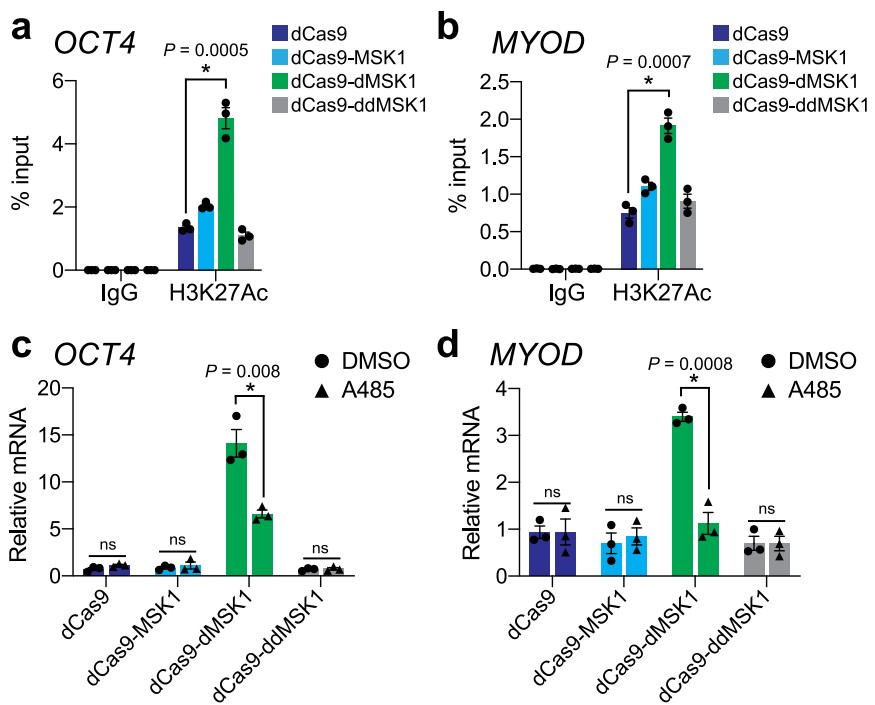

**Fig. 5 dCas9-dMSK1 influences H3K27 acetylation status at targeted human promoters. a, b** ChIP-qPCR enrichment for H3K27ac at the *OCT4* (panel **a**) or *MYOD* (panel **b**) promoters 72 h post-transfection of dCas9 or indicated dCas9-fusion protein and corresponding gRNAs. **c, d** RT-qPCR for *OCT4* (panel **c**) or *MYOD* (panel **d**) in the presence of DMSO or the CBP/p300 inhibitor A485 72 h post-transfection of dCas9 or indicated dCas9-fusion protein and corresponding gRNAs. 20 μM of A485 or an equal volume of DMSO was added to cells 12 h post-transfection. Two-sided *t*-test, *\*P < 0.05; n = 3 independent experiments for all panels; error bars, s.e.m.; ns, not significant. Source data are available in the Source data file.

identify genes that when synthetically overexpressed by dCas9-dMSK1, would result in resistance to the BRAF V600E inhibitor PLX-4720 (refs. [28,30]). Briefly, we stably transduced A375 melanoma cells with a dCas9-dMSK lentiviral expression vector and then transduced these cells with a previously developed CRISPR activation (CRISPRa) gRNA library[28] at a multiplicity of infection (MOI) of 0.2. Cells were then subjected to vehicle (DMSO) or

PLX-4720 treatment for 16 days, after which we used next-generation sequencing to recover gRNA sequences enriched in the PLX-4720 treated population relative to DMSO control-treated cells (Fig. 6a).

dCas9-dMSK1 was well-expressed and catalytically active in A375 cells (Supplementary Fig. 16), and in both experimental replicates, we observed increased enrichment of gRNAs in PLX-

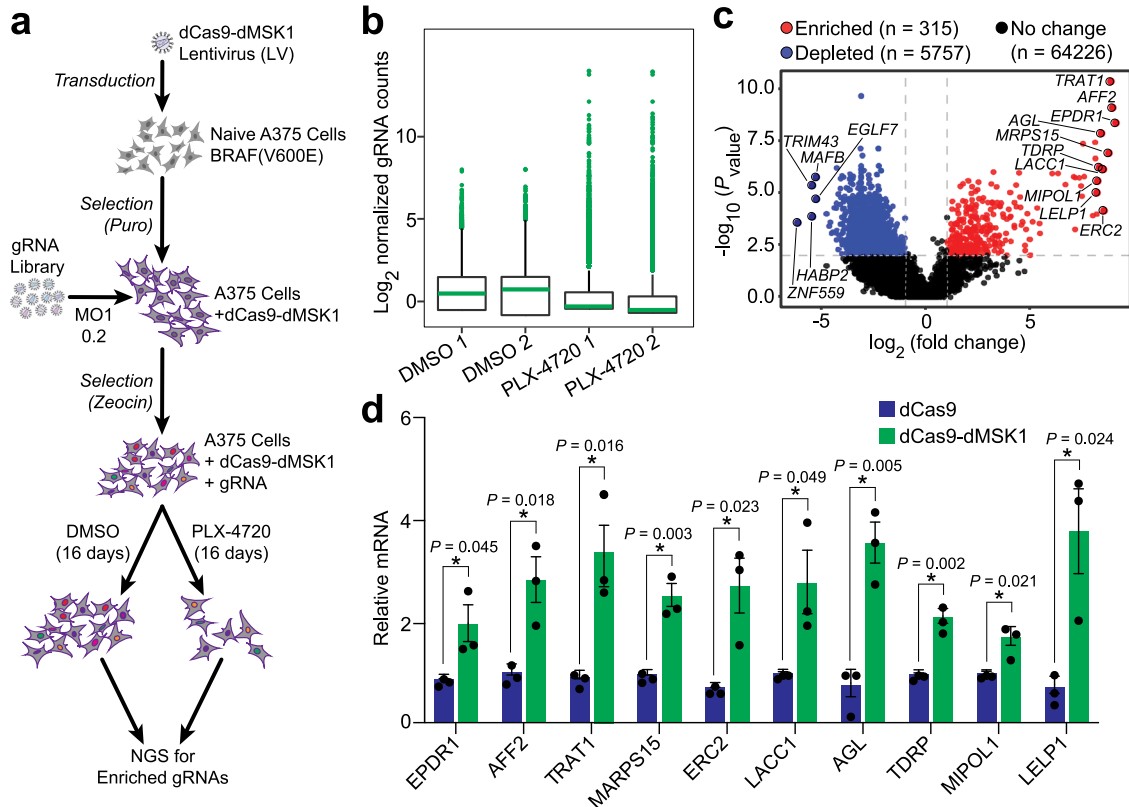

**Fig. 6 Genome-scale screening using dCas9-dMSK1 identifies mediators of BRAF V600E inhibitor resistance. a** Flow-chart of the genome-scale screening regime using dCas9-dMSK1. Puro; puromycin. **b** Box plots showing the distribution of gRNA frequencies after lentiviral transduction in DMSO or PLX-4720 treated A375 cells. In all box plots, the green horizontal line represents the median and the upper and lower bounds of each box correspond to the 25th and 75th percentiles of each sample, respectively. The upper and lower whiskers extend to the maximum and minimum values, respectively, within 1.5-fold of the interquartile range of the box bounds. Outliers are plotted as green dots. $n = 2$ independent experiments. **c** Volcano plot displaying gRNA counts in PLX-4720 versus DMSO treated A375 cells. Data were analyzed using robust rank aggregation (RRA) with a cutoff of $P$ value <0.01. Blue circles indicate depleted gRNAs, red circles indicate enriched gRNAs, and gray circles indicate gRNAs with no change. **d** mRNA levels of the top 10 most enriched genes in A375 cells stably expressing dCas9 or dCas9-dMSK1 when targeted with corresponding gRNAs identified from the screen. Two-sided $t$-test, $*P < 0.05$; $n = 3$ independent experiments for panel (**d**); error bars, s.e.m. Source data are available in the Source data file.

4720 treated samples relative to DMSO treated control cells (Fig. 6b). Based on the ratio of normalized gRNA counts in PLX-4720 vs. DMSO treated cells, there were 5757 significantly ($P$ value <0.01) depleted gRNAs (corresponding to 4889 targeted genes/promoters) and 315 significantly ($P$ value <0.01) enriched gRNAs (corresponding to 314 targeted genes/promoters) in the PLX-4720 treated cells (Fig. 6c, Supplementary Table 4). Eighteen of the 314 enriched genes were identified previously using the dCas9-SAM platform[28]. The differences in enriched gRNAs between dCas9-SAM and dCas9-dMSK1 could be due to the lower relative potency of dCas9-dMSK1 compared to dCas9-SAM (Supplementary Fig. 17), and/or unique effects associated with histone phosphorylation at the targeted promoters identified in this study.

Three of the top 10 genes identified as mediators of PLX-4720 resistance in our screen; *EPDR1*, *AFF2*, and *ERC2*, have been linked to the BRAF protein in colorectal cancer[31], autism spectrum disorders[32], and ganglioma[33], respectively. *MRPS15, TRAT1, LACC1, AGL, TDRP, MIPOL1,* and *LELP1* are mediators of PLX-4720 resistance that have not been previously reported. To validate our screening results, we targeted dCas9-dMSK1 (or a dCas9 control) to the promoters of *EPDR1, AFF2, ERC2, MRPS15, TRAT1, LACC1, AGL, TDRP, MIPOL1,* and *LELP1* using the gRNAs enriched from the screen (Fig. 6d). In each case, dCas9-dMSK1 significantly ($P$ value <0.05) increased the expression of each targeted gene. To confirm that the dCas9-

dMSK1-mediated induction of hits identified in our screen, we used dCas9-dMSK1 (and corresponding gRNAs) to induce the overexpression of *EPDR1* or *AFF2* (2 of the top hits from our screen) in A375 cells and then treated cells with either DMSO (control) or 3.5 µM PLX-4720 for 72 h. Microscopy analysis and MTT assays indeed revealed that dCas9-dMSK1-mediated upregulation of *EPDR1* or *AFF2* results in improved cell fitness and higher viability compared to dCas9 control-treated A375 cells when challenged with PLX-4720 (Supplementary Fig. 18). These results demonstrate that dCas9-dMSK1-mediated histone phosphorylation and gene activation are compatible with high-throughput gRNA screening approaches, and that programmable histone phosphorylation can be leveraged to uncover unique insights into human gene regulation and pathology.

In summary, we have built a CRISPR/Cas9-based epigenome editing tool, called dCas9-dMSK1, that permits writing histone phosphorylation at specific endogenous loci within native human chromatin. Using this capability, we have shown that histone phosphorylation plays a causal role in the activation of human promoters, and that H3S28ph is pivotal for this function. Furthermore, we have utilized dCas9-dMSK1 to demonstrate that functional crosstalk exists between histone phosphorylation and acetylation at endogenous human histones, and that this interplay is linked to the activation of gene expression from human promoters. Finally, we used dCas9-dMSK1 in combination with high-throughput genome-scale screening to uncover

genes involved in the development of therapeutic resistance. Our work expands the CRISPR/Cas9-based epigenome editing toolkit, clarifies the mechanistic role that histone phosphorylation plays in regulating human genes, and provides an efficient way to synthetically manipulate an important epigenomic modification that is ubiquitous across eukaryotes.

## Methods

**Cell culture and plasmid construction**. HEK293T cells (ATCC, CRL-11268), A549 cells (ATCC, CCL-185), and A375 cells (ATCC, CRL-1619) were cultured in Dulbecco's modified Eagle's medium (DMEM, Gibco, 31-053-028) supplemented with 10% FBS (Sigma, F2442) and 1% penicillin/streptomycin and maintained at 37 °C and 5% $CO_2$. The cloning backbone pLV-dCas9-p300-P2A-Puro (dCas9-P300, Addgene, 83889) has been described previously[29]. N-terminal MSK1 (1–1644 nt) was amplified from the pDONR223-RPS6KA5 (Addgene, 23758) and C-terminal MSK1 (1645–2406 nt) was synthesized as a gBlock gene fragment (Integrated DNA Technologies). The pLV-dCas9-MSK1-P2A-Puro construct (dCas9-MSK1) was created by subcloning full-length MSK1 into a BamHI-digested pLV-dCas9-p300-P2A-Puro backbone via NEBuilder HiFi DNA Assembly (NEB, E2621). The pLV-dCas9-dMSK1-P2A-Puro (dCas9-dMSK1) was generated by amplifying amino acids 42–801 of MSK1 from dCas9-MSK1 and then subcloning Δ41MSK1 into a BamHI-digested pLV-dCas9-p300-P2A-Puro backbone. The pLV-dCas9-ddMSK1-P2A-Puro (dCas9-ddMSK1) was created by amplification of dMSK1 from dCas9-dMSK1 in overlapping fragments with primer sets harboring the specified nucleic acid mutations (D195A and D565A), and then cloning this PCR fragment into the BamHI-digested pLV-dCas9-p300-P2A-Puro backbone using NEBuilder HiFi DNA Assembly (NEB, E2621). Protein sequences of all dCas9 constructs are shown in Supplementary Note 1. All gRNAs used for non-screening experiments were cloned into the pSPgRNA backbone (Addgene, 47108)[34]. All gRNA protospacer targets are listed in Supplementary Table 5. gRNA library plasmids[28] for genome-scale screening were purchased from Addgene (1000000057).

**Transfection**. Transient transfections were performed in 24-well plates using 375 ng of respective dCas9 expression vector and 125 ng of single gRNA vectors or equimolar pooled gRNA expression vectors. Plasmids were mixed with Lipofectamine 3000 (Invitrogen, L3000015) as per manufacturer's instruction. For ChIP-qPCR experiments, HEK293T cells were transfected in 15 cm dishes with Lipofectamine 3000 and 37.5 μg of respective dCas9 expression vector and 12.5 μg of equimolar pooled gRNA expression vectors as per manufacturer's instruction.

**Western blotting**. Twenty micrograms of protein was loaded for SDS-PAGE and transferred onto a PVDF membrane for Western blots. Primary antibodies (α-FLAG; Sigma-Aldrich, F1804; α-MSK1; Abcam, ab99412; α-MSK1S376ph; Abcam, ab32190; α-MSK1S212ph; Abcam, ab79499) were used at a 1:1000 dilution in 1X Tris Buffered Saline with 1% Casein (Bio-Rad, 1610782EDU). Secondary α-mouse HRP (Sigma-Aldrich, A6154) or α-rabbit HRP (Abcam, ab6721) were used at a 1:3000 dilution in 1X Tris Buffered Saline with 1% Casein (Bio-Rad, 1610782EDU). Membranes were exposed after addition of ECL (Bio-Rad, 170-5060). Tubulin was detected with hFAB™ Rhodamine Anti-Tubulin Primary Antibody (Bio-Rad, 12004166; 1:3000 dilution).

**Histone octamer in vitro kinase assay**. dCas9 and dCas9-fusion proteins were purified from transfected HEK293T cells using ANTI-FLAG® M2 Affinity Gel (Sigma, A2220) per manufacturer's instruction. Kinase assays were performed at 30 °C for 1 h in 30 μl reactions containing 500 ng of assembled human histone octamer, 150 nM of dCas9 or dCas9-fusion protein, 10 mM Tris-HCl, pH 8.0, 100 mM KCl, 10 mM MgCl₂, 0.1 M ATP and 1X phosphatase inhibitor (Thermo Scientific, 78425). The resulting reaction was suspended in SDS-PAGE sample buffer (Bio-Rad, 1610747), boiled at 95 °C for 10 min, separated by SDS-PAGE, and then transferred onto a PVDF membrane for Western blotting. After blocking with 1% casein for 1 h, phosphorylated histones were detected by immunoblotting with an appropriate phosphor-specific antibody at a 1:1000 dilution (α-H3S10ph; Abcam, 5176, or α-H3S28ph; Abcam, ab32388). Non-phosphorylated histones were detected using α-H3 (Abcam, ab1791; 1:1000 dilution). Secondary α-rabbit HRP (Abcam, ab6721; 1:3000 dilution) was used and membranes were exposed and imaged after the addition of ECL (Bio-Rad, 170-5060). Densitometry analysis was carried out using ImageJ.

**MSK1 knockout cell lines**. In total, 2 gRNAs (listed in Supplementary Table 5) targeting either exon 1 or exon 2 of human MSK1 were cloned into the Lenti-CRISPR v2 vector (Addgene, 52961). The resulting gRNA constructs were transiently transfected to HEK293T cells. Forty-eight hours post-transfection, cells were passaged and 1 μg/ml puromycin was added 3 h after plating. After 4 days of puromycin selection, cells were harvested, diluted 1000-fold, and replated in 15-cm plates with complete medium supplemented with 1 μg/ml puromycin. Two weeks later, single colonies were picked and cultured within 96-well plates and after 1 week of growth, cells were passaged into 24-well plates. MSK1 knockout lines

were confirmed by western blot analysis with α-MSK1 (Abcam, ab99412; 1:1000 dilution) and secondary α-rabbit HRP (Abcam, ab6721; 1:3000 dilution). The LentiCRISPR v2 empty vector was used as a control in WT HEK293T cells.

**RNA sequencing**. RNA sequencing (RNA-seq) was performed in duplicate for each experimental condition. RNA was isolated from transfected cells using the RNeasy Plus mini kit (Qiagen, 74136). RNA-seq libraries were constructed using the TruSeq Stranded Total RNA Gold (Illumina, RS-122-2303). The qualities of RNA-seq libraries were verified using the Tape Station D1000 assay (Tape Station 2200, Agilent Technologies) and the quantities of RNA-seq libraries were checked again using real-time PCR (QuantStudio 6 Flex Real time PCR System, Applied Biosystem). Libraries were normalized and pooled and then 75 bp paired-end reads were sequenced on the Hiseq3000 platform (Illumina). Reads were aligned to the Hg38 transcriptome using HISAT2(2.1.0)[35]. Transcript abundance was calculated using feature Counts from the subread package (v2.0.0)[36], and differential expression was determined in R studio (v1.2.13) using the DESeq2 (v1.28.1) analysis package with default parameters[37]. Gene ontology analysis was performed using DAVID Functional Annotation Bioinformatics Microarray Analysis[38] at https://david.ncifcrf.gov/. All RNA-seq reads have been uploaded to the NCBI single read archive and made publicly available upon publication acceptance.

**Reverse-transcription quantitative PCR (RT-qPCR)**. RNA was isolated from transfected cells using the RNeasy Plus mini kit (Qiagen, 74136) and 1 μg of purified RNA was used as template for cDNA synthesis (Bio-Rad, 1725038). Real-time quantitative PCR was performed using SYBR Green (Bio-Rad, 1725275) and a CFX96 Real-Time PCR Detection System with a C1000 Thermal Cycler (Bio-Rad, 1855195). Baselines were subtracted using the baseline subtraction curve fit analysis mode and thresholds were automatically calculated using the Bio-Rad CFX Manager software version 2.1. Results are expressed as fold change above cells transfected with an empty vector plasmid (Addgene, 47108) after normalization to GAPDH expression using the ΔΔCt method. All qPCR primers and conditions are listed in Supplementary Table 6.

**ChIP-qPCR**. HEK293T cells were co-transfected with indicated dCas9-fusion expression vectors and gRNA constructs in 15 cm plates in biological triplicates for each condition tested. Cells were cross-linked for 10 min at RT using 1% formaldehyde (Sigma F8775-25ML) and then the reaction was stopped by the addition of glycine to a final concentration of 125 mM. Cells were harvested and washed with ice-cold 1X PBS and suspended in Farnham lysis buffer (5 mM PIPES pH 8.0, 85 mM KCl, 0.5% NP-40) supplemented with 1X protease and phosphatase inhibitor (Thermo Scientific, 78426). Cells were then pelleted and resuspended in RIPA buffer (1X PBS, 1% NP-40, 0.5% sodium deoxycholate, 0.1% SDS) supplemented with 1X protease and phosphatase inhibitor. Approximately 2.5e7 cells were used for each H3S10ph and H3S28ph ChIP experiment. Chromatin in RIPA buffer was sheared to a median fragment size of 250 bp using a Bioruptor XL (Diagenode). Two micrograms of each antibody (α-FLAG; Sigma-Aldrich, F1804, Mouse IgG; Abcam, ab18413, Rabbit IgG; Abcam, ab171870, α-H3S10ph; Abcam, ab17246, α-H3S28ph; Abcam, ab32388, α-H3K4me3; Abcam, ab8580, or α-MSK1; Abcam, ab99412) was incubated with 50 μl Sheep anti-Rabbit or Mouse IgG magnetic beads (Life Technologies, 11203D or 11202D) for 16 h at 4 °C, respectively. Antibody-linked magnetic beads were washed three times with PBS/BSA buffer (1X PBS and 5 mg/ml BSA) and then sheared chromatin was incubated with corresponding antibody-linked magnetic beads at 4 °C overnight and then washed five times with LiCl IP wash buffer (100 mM Tris pH 7.5, 500 mM LiCl, 1% NP-40, 1% sodium deoxycholate). Cross-links were then reversed via overnight incubation at 65 °C and DNA was purified using QIAquick PCR purification kit (Qiagen, 28106) for ChIP-qPCR or ChIP-seq. Input DNA was prepared from ~1.0e6 cells. Ten nanograms of DNA was used for subsequent qPCR reactions using a CFX96 Real-Time PCR Detection System with a C1000 Thermal Cycler (Bio-Rad, 1855195). Baselines were subtracted using the baseline subtraction curve fit analysis mode and thresholds were automatically calculated using the Bio-Rad CFX Manager software version 2.1. The ChIP-qPCR data is plotted relative to percent input. All ChIP-qPCR primers and conditions are listed in Supplementary Table 6.

**ChIP sequencing**. ChIP-sequencing (ChIP-seq) libraries were prepared using the KAPA Hyper Prep Kit (Roche) following the manufacturer's protocol. Briefly, 3 ng of fragmented DNA was used to repair the ends before ligation to a diluted NEXTflex DNA adapter (PerkinElmer). The adaptor-ligated DNA was then enriched using 10 cycles of PCR (1 cycle at 98 °C for 45 s; 10 cycles at 98 °C for 15 s, 60 °C for 30 s, 72 °C for 30 s; and 1 cycle at 72 °C for 1 min). DNA was purified using AMPure XP beads (Beckman Coulter) after adaptor ligation and PCR enrichment, and the libraries were run on a 2200 Tape Station (Agilent Technologies) for quality control. A KAPA Library Quantification Kit (KAPA Biosystems) was utilized to quantify the libraries for pooling, and a final concentration of 1.5 nM was loaded onto an Illumina cBOT for cluster generation before sequencing on an Illumina HiSeq3000 for sequencing using a Single Read 50 bp run. For ChIP-seq data analysis, reads were aligned to the hg38 reference genome using Bowtie2 (v2.3.4.2)[39] and duplicates were removed using the rmdup tool from SAMtools (v1.9)[40]. Reads from each duplicate for each condition were combined, and peaks

were called using MACS2 (v2.1.2.1)[41]. Resulting peaks from each condition with a $q$-value $\leq 0.01$ were merged using the mergeBed tool from BEDTools (v2.27.1)[42]. Read numbers in peaks were estimated using feature Counts from the subread package (v2.0.0)[36]. The difference in binding was assessed with R studio (v1.2.13) and DESeq2 (v1.28.1)[37] with an FDR cutoff of $\leq 0.01$.

**Re-ChIP**. For re-ChIP experiments[43], ~5e7 cells were used for the first ChIP and chromatin in RIPA buffer was sheared to a minimal fragment size of 250 bp using a Bioruptor XL (Diagenode). The first ChIP was performed by incubation of sheared chromatin with corresponding antibody-linked magnetic beads at 4 °C overnight. Samples were washed three times with Re-ChIP wash buffer (50 mM Tris-HCl pH 8.0, 500 mM NaCl, 0.1% SDS, 1% NP-40, 2 mM EDTA) and then chromatin-antibody complexes were eluted in Re-ChIP elution buffer (10 mM Tris-HCl pH 8.0, 2% SDS, 15 mM DTT) at 37 °C for 30 min and subsequently diluted 1/20 in ChIP dilution buffer (16.7 mM Tris-HCl pH 8.0, 167 mM NaCl, 1.2 mM EDTA, 1.1% Triton X-100, 0.01% SDS). The second antibody-linked-magnetic beads were added and incubated overnight on a rotary shaker at 4 °C. The second-round chromatin-antibody complexes were captured, washed, and eluted similarly and then DNA was purified for qPCR as above.

**Lentivirus production**. One day before transfection, HEK293T cells were seeded at ~40% confluency in a 10-cm plate. The next day cells were transfected at ~80–90% confluency. For each transfection, 10 µg of plasmid containing the vector of interest, 10 µg of pMD2.G (Addgene, 12259), and 15 µg of psPAX2 (Addgene, 12260) were transfected using calcium phosphate. Five hours post-transfection the media was changed. Supernatant was harvested 24 and 48 h post-transfection and filtered with a 0.45-µm PVDF filter (Millipore, SLGVM33RS), and then virus was concentrated using Lenti-X™ Concentrator (Takara, 631232), aliquoted and stored at −80 °C. Lentiviral titers were measured by the Lenti-X™ qRT-PCR Titration Kit (Takara, 631232).

**Lentiviral transduction for stable cell lines**. A375 cells were transduced with lentiviruses encoding dCas9 or dCas9-dMSK1 in 6-well plates at a MOI of 10. Briefly, 1e6 cells in 2 ml of media supplemented with 8 µg per ml polybrene (Sigma, TR-1003-G) were added to each well. Then, 48 h post-transduction, cells were passaged and 1 µg/ml puromycin was added 3 h after plating. Media was exchanged 2 days post-transduction and cells were passaged every other day starting 4 days after initial replating. Puromycin selection was maintained for a total of 7 days.

**CRISPR screening assay**. A375 cells stably transduced with dCas9-dMSK1 were transduced with gRNA libraries at a MOI of 0.2, with a minimal representation of 500 transduced cells per gRNA. Cells were maintained at >500 cells per guide during subsequent passaging. At 7 days post-transduction cells were split into DMSO and PLX-4720 conditions (2 µM PLX-4720 dissolved in DMSO; Selleckchem). Cells were then passaged every 3 days for a total of 16 days of drug or vehicle treatment. Greater than 500 cells per guide were harvested at 23 days post-transduction (16 days post-treatment) for genomic DNA extraction using the Qiagen Quick-gDNA midi kit (Qiagen, 13343). Eight micrograms of genomic DNA was used as a template across eight 100 µl PCR reactions to amplify gRNAs using Q5 hot start polymerase (NEB, M0493L). Amplification was carried out as per manufacturer's instructions, using 25 cycles at an annealing temperature of 60 °C with the following primers:

Fwd: 5′-AATGATACGGCGACCACCGAGATCTACACNNNNNNNNNACACT CTTTCCCTACACGACGCTCTTCCGATCTGGACTATCATATGCTTACCGTA ACTTG (8-bp index read barcode indicated by italics); Rev: 5′-CAAGCAGAAG ACGGCATACGAGATNNNNNNNNNGTGACTGGAGTTCAGACGTGTGCTCT TCCGATCTGCCAAGTTGATAACGGACTAGCCTT (8-bp index read barcode indicated by italics).

Amplified libraries were purified using the QIAquick Gel Extraction Kit (Qiagen, 28706). Each sample was quantified after purification with the Qubit dsDNA High Sensitivity assay kit (Thermo Fisher, Q32854). Samples were pooled and sequenced using the MiSeq platform (Illumina). All reads from gRNA screening will be uploaded to the NCBI single read archive and made publicly available upon publication acceptance.

**Data analysis**. Samples were analyzed using the MAGECK-VISPR algorithm[44] (quality control, modeling, and visualization of CRISPR screens with MAGeCK-VISPR) using Robust Rank Aggregation (RRA) default parameters. NGS data were de-multiplexed using uniquely indexed reads. Guide counts were determined based on perfectly matched sequencing reads only. For each condition, guide counts were normalized to the total number of counts per condition, and log2 counts were calculated based on these values. Ratios of counts between conditions were calculated as log2 ((count 1 + 1)/(count 2 + 1)) based on normalized counts. Associated box plots and gRNA expression change volcano plots were created using ggplot2 (https://ggplot2.tidyverse.org).

**Validation of BRAF(V600E) inhibitor (PLX-4720) candidate genes**. Ten of the most enriched candidate gRNAs were cloned into Lenti_sgRNA (MS2) zeo

backbone (Addgene, 61427). Lentiviruses were then produced, harvested, and transduced into A375 cells stably transduced with dCas9 control or dCas9-dMSK1. Cells were then put under 200 µg/ml Zeocin selection for 7 days, after which mRNA was harvested and measured using RT-qPCR as described above. All gene expression levels in A375 cells were normalized to A375 cells stably transduced with dCas9. Stably transduced cell lines with an *EPDR1* or *AFF2* gRNA were treated with 3.5 µM PLX-4720 for 72 h and cell morphology was observed using microscopy and cell viability was determined using MTT assays.

**MTT viability assay**. Cells were plated at 2e3 per well in 96-well plates and incubated at 37 °C for 12 h before treatment with either DMSO or PLX-4720, in 200 µl. MTT reagent (3-(4,5-dimethythiazol-2-yl)-2,5-diphenyl tetrazolium bromide; Cayman, 21795-1) was then added to a final concentration of 0.5 mg/ml. The reaction was incubated at 37 °C for 3 h and then media and MTT reagent were removed. Fifty microliters of DMSO was then added and cells were incubated at 37 °C for 30 min to 2 h, until cells had lysed and purple crystals had dissolved. Medium only wells were used as blanks. Absorbance was then measured at 570 nanometers and then cell viability was calculated using the formula: $(\text{abs}_{\text{PLX4720}} − \text{abs}_{\text{Blank}})/(\text{abs}_{\text{DMSO}} − \text{abs}_{\text{Blank}}) \times 100$.

**Reporting summary**. Further information on research design is available in the Nature Research Reporting Summary linked to this article.

## Data availability
RNA sequencing, ChIP sequencing, and CRISPR screening data are deposited in the NCBI Gene Expression Omnibus repository under GEO accession number GSE156381. All dCas9-based MSK1 fusion protein variants are available through Addgene and all reagents are available from the authors upon request. Any other relevant data are available from the authors upon reasonable request. Source data are provided with this paper.

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

## Acknowledgements

This work was supported by a Cancer Prevention & Research Institute of Texas (CPRIT) Award to I.B.H. (RR170030) and partly by a CPRIT Core Facility Support Award (RP170002). We thank J.-J. Shen and the MD Anderson Cancer Center Science Park NGS core, David Zhang, and Zhang lab members at Rice University for assistance with next-generation sequencing. We thank all members of the Hilton laboratory for helpful discussions.

## Author contributions

J.L. and I.B.H. conceived the study. J.L., B.M., and I.B.H. designed the experiments. J.L. carried out experiments with the assistance of B.M., M.E., J.G., K.W., and P.K. J.L. and I.B.H. wrote the manuscript with input from all authors and input from Daniel Brenner and Maxwell Hunt.

## Competing interests

The authors declare no competing interests.
