## [Peer Review File · Nature Communications]

Reviewers' Comments:

Reviewer #1:

Remarks to the Author:

The authors have generated a novel dCas9-dMSK construct to target H3S10ph or H3S28ph to specific sites. This approach of targeting MSK is not new and the authors have not done a very thorough review of the literature in not citing the article PMID: 21282660. The authors statement "however no tools have been created that permit the site-specific modification of histone phosphorylation within native chromatin" is therefore false. The tool is interesting and should be presented in a Methods but certainly not in Nature Communications. The results are interesting but do not present new information about the mechanisms involved.

Comments

1. The labelling under Fig 1d is wrong. In this fig dCas9-ddMSK1 is showing activity but it is catalytically dead.
2. Multiple studies have shown that MSK phosphorylates either S10 or S28 but not both (PMID: 15870105, 28077620, 15735677). The authors need to apply sequential ChIP assays (re-ChIP).
3. As the increase in targeted H3S28ph and H3K27ac occur at promoters, the authors should also study the status of H3K4me3 at the promoter and at the beginning of the target genes.
4. Studies have shown that inhibition of CBP/p300 prevented H3S28ph by MSK. The authors should test this to know more about the relationship between MSK and CBP/p300.
5. The authors should present ChIP-Seq for H3S10ph and H3S28ph when targeting dCas9-dMSK1 to multiple genomic sites.

Reviewer #2:

Remarks to the Author:

MSK1 is an important effector kinase of MAP kinase pathways and phosphorylates, besides transcription factors, histone H3 at serines 10 and 28. The presence of MSK1 (or MSK2) has been previously shown to correlate with histone H3 phosphorylation and gene activation of many stimulus-responsive genes. In this manuscript Li et al. used a CRISPR/Cas9-based targeting approach to test whether recruitment of MSK1 or a hyperactive MSK1 variant is able to activate target genes and non-natural target genes. The authors show that in particular the targeting of hyperactive MSK1 results in histone H3 phosphorylation and induced gene expression. In this context, the H3S28ph mark seems to be more important than H3S10ph. In addition, also other genes that are not activated in the absence of MSK1 can be induced by active MSK1. Importantly, the recruitment of MSK1 to a specific promoter region and not to the enhancer (in case of OCT4) is required to induce the transcriptional activation. Finally, the authors successfully performed a genome-wide screen with the hyperactive mutant to identify genes which mediate resistance to the BRAF V600E inhibitor PLX-4720 in melanoma cells.

This is an elegant study based on well-documented experiments of high technical quality. The results are novel, highly relevant and important not only for the epigenome community but also for cell signaling and cancer research. The authors could also discuss that this new tool might be instrumental in the analysis of regulatory mechanisms controlling stimulus-induced transcription by studying the role of CREB/ATF1 phosphorylation and recruitment of reader proteins in response to MSK1 targeting.

Comments:

Page 4, line 86: These genes are bona fide target genes because, as the authors state later, the downregulation upon MSK1 KO could be indirect. Do the authors have evidence for binding of MSK1 to the regulatory regions of these genes from the literature or their own experiments?

Page 9, line 194: The authors validate the induction of target genes in response to MSK1 recruitment. It would be important to show that overexpression of one or two targets indeed mediates resistance to the BRAF inhibitor.

Page 4, line 73: typo

AUTHORS' RESPONSE TO REVIEWER COMMENTS

Reviewer #1 (Remarks to the Author):

The authors have generated a novel dCas9-dMSK construct to target H3S10ph or H3S28ph to specific sites. This approach of targeting MSK is not new and the authors have not done a very thorough review of the literature in not citing the article PMID: 21282660. The authors statement "however no tools have been created that permit the site-specific modification of histone phosphorylation within native chromatin" is therefore false. The tool is interesting and should be presented in a Methods but certainly not in Nature Communications. The results are interesting but do not present new information about the mechanisms involved.

Author Response: We are sincerely grateful to Reviewer #1 for their time and expertise, and we appreciate their thoughtful and incisive comments. We have addressed the helpful comments from Reviewer #1 below, and in so doing, we feel that the input from Reviewer #1 has improved our manuscript and strengthened our findings. Changes based on Reviewer input in the revised main text and supplemental materials are colored red for clarity.

*We thank Reviewer #1 for identifying our oversight in failing to cite seminal work to modulate histone phosphorylation at specific endogenous loci by Lau and Cheung (2011; PMID 21282660) in our initial submission. We have corrected this oversight in our resubmitted manuscript by citing this publication, modifying our language, and by emphasizing the importance of PMID 21282660 to understanding the function of MSK1 and endogenous histone phosphorylation.

We agree that by building a fusion protein consisting of the DNA binding domain of the NF1 transcription factor and full length MSK1 in PMID 21282660, Lau and Cheung created a tool that permits locus-specific modification of histone phosphorylation within native chromatin. Notably Lau and Cheung used this fusion protein to direct MSK1 to cognate NF1 binding sites in HEK293T cells and elegantly demonstrated that the recruitment of full length MSK1 to native NF1 binding sites induced phosphorylation of nearby histone H3S10 and H3S28 residues and induced gene expression levels at two endogenous human genes (*c-fos* and *α -globin*). Lau and Cheung also showed that these effects were functionally connected to PRC2 (i.e., EZH2) displacement and increased local histone acetylation levels (i.e., H3K27ac).

Although the recruitment method developed and applied by Lau and Cheng is powerful and mechanistically useful for endogenous loci harboring NF1 binding sites, the strategy presumably results in full length MSK1 recruitment to all loci harboring NF1 binding sites, and therefore is not precisely targetable to a single specific locus in human cells. Moreover, the strategy employed by Lau and Cheung cannot dissect the role of MSK1-driven histone phosphorylation at loci that do not harbor NF1 binding sites. In contrast, because our approach leverages the programmability of dCas9, we have demonstrated that we can target full length MSK1 or a hyperactive variant of full length MSK1 (dMSK1) to virtually any endogenous human locus. Furthermore, in our revised manuscript we have also shown that this targeting is highly specific across the human genome using both ChIP- and RNA-seq (Figures 3c and 3d, respectively). These technological advances will expand the utility for other researchers/research groups interested in understanding and synthetically harnessing endogenous histone phosphorylation.

*We appreciate the concern of Reviewer #1 surrounding the publication of our manuscript in *Nature Communications*. We respectfully point the reviewer to the following selected letters and articles that have been recently published in *Nature Communications* based upon the development and application of dCas-based epigenome editing tools.

-2016; PMID: 27506838; Cano-Rodriguez et al.; *Writing of H3K4Me3 overcomes epigenetic silencing in a sustained but context-dependent manner*

-2016; PMID: 27694915; Ma et al.; *Integration and exchange of split dCas9 domains for transcriptional controls in mammalian cells*

-2017; PMID: 28497787; Kwon et al.; *Locus-specific histone deacetylation using a synthetic CRISPR-Cas9-based HDAC*

-2017; PMID: 28541304; Gander et al.; *Digital logic circuits in yeast with CRISPR-dCas9 NOR gates*

-2017; PMID: 28695892; Lei et al.; *Targeted DNA methylation in vivo using an engineered dCas9-MQ1 fusion protein*

-2017; PMID: 28703221; Morgan et al.; *Manipulation of nuclear architecture through CRISPR-mediated chromosomal looping*

-2017; PMID: 28916764; Braun et al.; *Rapid and reversible epigenome editing by endogenous chromatin regulators*

-2017; PMID: 29084946; Kleinjan et al.; *Drug-tunable multidimensional synthetic gene control using inducible degron-tagged dCas9 effectors*

-2017; PMID: 29158476; Hao et al.; *Programmable DNA looping using engineered bivalent dCas9 complexes*

-2018; PMID: 29426832; Galonska et al.; *Genome-wide tracking of dCas9-methyltransferase footprints*

-2018; PMID: 29700298; Thakore et al.; *RNA-guided transcriptional silencing in vivo with S. aureus CRISPR-Cas9 repressors*

-2018; PMID: 29950558; Dong et al.; *Synthetic CRISPR-Cas gene activators for transcriptional reprogramming in bacteria*

-2018; PMID: 29980666; Weltner et al.; *Human pluripotent reprogramming with CRISPR activators*

-2018; PMID: 30158531; Xu et al.; *High-fidelity CRISPR/Cas9- based gene-specific hydroxymethylation rescues gene expression and attenuates renal fibrosis*

-2019; PMID: 31541098; Taghbalout et al.; *Enhanced CRISPR-based DNA demethylation by Casilio-ME-mediated RNA-guided coupling of methylcytosine oxidation and DNA repair pathways*

-2020; PMID: 31980609; Li et al.; *Interrogation of enhancer function by enhancer-targeting CRISPR epigenetic editing*

-2020; PMID: 32001704; Krawczyk et al.; *Rewiring of endogenous signaling pathways to genomic targets for therapeutic cell reprogramming*

-2020; PMID: 32415193; Bhokisham et al.; *A redox-based electrogenetic CRISPR system to connect with and control biological information networks*

-2020; PMID: 32488086; Santos-Moreno et al.; *Multistable and dynamic CRISPRi-based synthetic circuits*

-2020; PMID: 32561716; Chen et al.; *Repurposing type I-F CRISPR-Cas system as a transcriptional activation tool in human cells*

Each of these studies above markedly expanded the programmability and utility of available tools and substantially advanced and broadly impacted relevant fields. In many of the publications referenced above, prior foundational work using other DNA binding scaffolds, such as Gal4 systems, ZFs, and/or TALEs, set the precedent for the effector domains and/or conceptual strategies used. We note further, that despite the fundamental importance of histone phosphorylation, our new technology is the first dCas-based tool that permits the ability to engender this mark. In addition, we have shown that dCas9-dMSK1 is robust at catalyzing histone phosphorylation *in vitro* and in cells, that it is highly specific in human cells on a genome-wide scale, and that the tool can be used in CRISPR screening approaches to uncover novel drivers of human pathology. We anticipate that these advances will be broadly useful for both basic and translational researchers, and that our findings will have impacts across scientific disciplines. Therefore, given these collective considerations, we strongly believe that our manuscript merits publication in *Nature Communications*.

*We respectfully disagree with Reviewer #1 that our results “do not present new information about the mechanisms involved”. We have briefly summarized the novel mechanistic insights into MSK1 biology and the role of histone phosphorylation from our work below. In addition, our findings strongly support and validate prior findings. Importantly, the ability to mechanistically dissect histone phosphorylation at specific and diverse endogenous human loci would not be possible without our new technology. We anticipate, and hope, that this new technology will be very useful for any researchers interested in understanding and repurposing endogenous histone phosphorylation.

1. We have identified human genes that are dysregulated upon loss of MSK1 using RNA-seq and transcriptome-wide analysis (**Figure 2a, Supplementary Figures 1c and 3**), potentially paving for the exploration of new drug targets downstream of MSK1 (PMID: 27768872). Although corresponding data exist in murine cells (GSE98751, GSE89141, and GSE62659), to our knowledge, this is the first comparative transcriptome-wide analysis between WT and MSK1 KO human cells.

2. We have shown that the truncated dMSK1 (lacking the N-terminal inhibitory domain; PMID: 20089855) is more potent than full length MSK1 at catalyzing H3S28ph both *in vitro* (**Figure 1**) and at diverse loci within native human chromatin (**Figures 2d, 4a, and 4b**). Further, at the loci we have tested in HEK293T cells herein, we have shown that the localization of full length MSK1 is insufficient to activate gene expression in the absence of other factors (e.g., the NF1 DBD; **Figures 2b, 3a, 3b, Supplementary Figures 12 and 14**).

3. We have observed that human enhancers appear to be relatively insensitive to the functional effects of histone phosphorylation (**Supplementary Figures 7 and 8**), and instead that histone phosphorylation, at least in our studies, has more functional potency at human promoters.

4. We have shown that histone phosphorylation-driven gene activation from promoters appears to be dose responsive (**Supplementary Figure 13**).

5. We have tested the effects of histone phosphorylation-driven gene expression at thousands of endogenous promoters in the human genome and shown that histone phosphorylation at promoters can result in acquired chemotherapeutic resistance (**Figure 6**). Notably, our results have only partial overlap (18 of 314; ~5.7%) with previous genome-scale screening studies using different dCas9-based activators (PMID: 25494202), and in addition to identifying established interactions (*EPDR1*, *AFF2*, and *ERC2*; PMIDs: 30360391, 21129364, and 29880043), and proving that their upregulation (*EPDR1* or *AFF2*) mediates PLX-4720 resistance (**Supplementary Figure 18**), we also found novel drivers of PLX-4720 resistance. Together, these data suggest that different regulatory mechanisms (e.g. histone phosphorylation vs. direct transcription factor-driven activation) can result in different phenotypic outcomes independent of absolute levels of gene expression *per se*.

6. Our results have validated and strengthened the evidence from other groups that:

-Phosphorylation of H3S28 appears to functionally predominate over phosphorylation of H3S10 in terms of gene activation from endogenous human promoters, suggesting that H3S28ph and H3S10ph are not functionally equivalent (e.g. PMIDs: 21282660, and 27768872).

-Endogenous H3S28ph displays functional crosstalk with H3K27ac and that targeted dMSK1-mediated gene activation is particularly sensitive to p300/CBP activity (21282660, 20864036, 27768872, and 25135956).

- H3S28ph and H3S10ph do not appear to co-occupy the same H3 histone tail (PMIDs: 15870105, 28077620, and 15735677).

Comments

1. The labelling under Fig 1d is wrong. In this fig dCas9-ddMSK1 is showing activity but it is catalytically dead.

Author Response: We thank Reviewer #1 for pointing out this discrepancy. We have re-quantified the data in **Figure 1d** to correct this inconsistency and have provided the original uncropped source data for all Western blots for clarity.

2. Multiple studies have shown that MSK phosphorylates either S10 or S28 but not both (PMID: 15870105, 28077620, 15735677). The authors need to apply sequential ChIP assays (re-ChIP).

Author Response: We agree and appreciate this important insight from Reviewer #1. We have performed re-ChIP (**Figures 4c and 4d**), and indeed our results indicate that H3S10ph and H3S28ph do not appear to coexist. These findings corroborate prior work (PMIDs: 15870105, 28077620, 15735677, 20129940, and 21282660) and further support the predominant role of H3S28ph relative to H3S10ph in promoter-driven gene expression. In addition to adding our updated data, we have also included appropriate text and cited the aforementioned manuscripts.

3. As the increase in targeted H3S28ph and H3K27ac occur at promoters, the authors should also study the status of H3K4me3 at the promoter and at the beginning of the target genes.

Author Response: We are grateful to Review #1 for this excellent advice. We have measured the levels of H3K4me3 at the promoter regions of *OCT4* and *MYOD* after targeting with dCas9-MSK1, dCas9-dMSK1, and dCas9/dCas9-ddMSK1 controls using ChIP-qPCR (**Supplementary Figure 11**). Surprisingly, and in contrast to previous reports (PMID: 27895715), we did not observe increased H3K4me3 enrichment coincident with dCas9-dMSK driven histone phosphorylation and gene activation. We note, that as a synthetic, programmable dCas9-based histone kinase, dCas9-dMSK1 may not work in exactly the same way, as natural full-length WT MSK1 in human cells.

4. Studies have shown that inhibition of CBP/p300 prevented H3S28ph by MSK. The authors should test this to know more about the relationship between MSK and CBP/p300.

Author Response: We agree and thank Reviewer #1 for this helpful suggestion. We targeted the *OCT4* or *MYOD* promoters with dCas9-MSK1, dCas9-dMSK1, and dCas9/dCas9-ddMSK1 controls in HEK293T cells while simultaneously inhibiting CBP/p300 using the potent and specific CBP/p300 catalytic inhibitor A485 (**Figure 5c and 5d**). Indeed, treatment of cells with A485 significantly ($P < 0.05$) reduced the dCas9-dMSK1 mediated mRNA levels of *OCT4* and *MYOD* relative to DMSO treated cells. These new data strongly implicate a functional relationship between CBP/p300 and dCas9-dMSK1 gene activation at human promoters.

5. The authors should present ChIP-Seq for H3S10ph and H3S28ph when targeting dCas9-dMSK1 to multiple genomic sites.

Author Response: We appreciate this excellent suggestion from Reviewer #1. We performed ChIP-seq to establish the genome wide binding specificity (using α -FLAG; **Figure 3c**) and the genome wide precision of H3S28ph deposition (using α -H3S28ph; **Supplementary Figure 9a**) of dCas9-dMSK1 when targeted to the *OCT4* promoter in HEK293T cells. We focused on H3S28ph as opposed to H3S10ph because our data and previous reports (e.g., PMIDs: 21282660 and 27768872) suggest that H3S28ph functionally predominates over H3S10ph in terms of endogenous human promoter activation. Our FLAG ChIP-seq results demonstrate that dCas9-dMSK1 binding is highly specific to targeted sites (**Figures 3c and Supplementary Figure 9b**).

Despite observing increased H3S28ph enrichment subsequent to targeted dCas9-dMSK1 to the OCT4 promoter in both ChIP-seq reads (**Supplementary Figure 9b**) and using ChIP-qPCR (**Figure 4a**), we did not identify significant H3S28ph ChIP-seq peaks on a genome-wide scale (FDR < 0.05). We suspect that the inability to detect significant H3S28ph enrichment using ChIP-seq may be due to a relatively narrow signal to noise ratio for H3S28ph in ChIP-seq experiments. In support of this hypothesis, it has been shown (see PMID: 21934668) that antibodies that display ≤ 5 -fold enrichment in ChIP-qPCR assays often fail in ChIP-seq experiments, despite being commercially marketed as “ChIP grade” (i.e., Anti-Histone H3 phospho S28 antibody - ChIP Grade; ab32388). In further support of this hypothesis, dCas9-dMSK1 induced H3S28ph enrichment ≤ 5 -fold relative to dCas9 control as measured by ChIP-qPCR at all loci we tested (**Figures. 2d, 4a, and 4b**). Nevertheless, our ChIP-qPCR results demonstrate that dCas9-dMSK1 mediates marked histone phosphorylation (both H3S10ph and H3S28ph) at target sites, and our ChIP-seq experiments demonstrate that the binding of dCas9-dMSK1 is highly specific, and finally that global H3S28ph levels are not dramatically altered upon dCas9-dMSK1 targeting.

Reviewer #2 (Remarks to the Author):

MSK1 is an important effector kinase of MAP kinase pathways and phosphorylates, besides transcription factors, histone H3 at serines 10 and 28. The presence of MSK1 (or MSK2) has been previously shown to correlate with histone H3 phosphorylation and gene activation of many stimulus-responsive genes. In this manuscript Li et al. used a CRISPR/Cas9-based targeting approach to test whether recruitment of MSK1 or a hyperactive MSK1 variant is able to activate target genes and non-natural target genes. The authors show that in particular the targeting of hyperactive MSK1 results in histone H3 phosphorylation and induced gene expression. In this context, the H3S28ph mark seems to be more important than H3S10ph. In addition, also other genes that are not activated in the absence of MSK1 can be induced by active MSK1. Importantly, the recruitment of MSK1 to a specific promoter region and not to the enhancer (in case of OCT4) is required to induce the transcriptional activation. Finally, the authors successfully performed a genome-wide screen with the hyperactive mutant to identify genes which mediate resistance to the BRAF V600E inhibitor PLX-4720 in melanoma cells. This is an elegant study based on well-documented experiments of high technical quality. The results are novel, highly relevant and important not only for the epigenome community but also for cell signaling and cancer research. The authors could also discuss that this new tool might be instrumental in the analysis of regulatory mechanisms controlling stimulus-induced transcription by studying the role of CREB/ATF1 phosphorylation and recruitment of reader proteins in response to MSK1 targeting.

Author Response: We appreciate the perceptive advice and insightful suggestions from Reviewer #2 and thank them for their time, knowledge, and supportive comments in thoughtfully reviewing our manuscript. We feel that the input from Reviewer #2 has improved our manuscript and bolstered the findings and impact. We have addressed each constructive comment from Reviewer #2 below. Changes based on Reviewer input in the revised main text and supplemental materials are colored red for clarity.

*We agree with Reviewer #2 and thank them for the advice to discuss that dCas9-dMSK1 might be instrumental in the analysis of regulatory mechanisms controlling stimulus-induced transcription by studying the role of CREB/ATF1 phosphorylation and recruitment of reader proteins in response to MSK1 targeting. We have added language surrounding the utility of dCas9-dMSK1 in this important research space.

Comments:

Page 4, line 86: These genes are bona fide target genes because, as the authors state later, the downregulation upon MSK1 KO could be indirect. Do the authors have evidence for binding of MSK1 to the regulatory regions of these genes from the literature or their own experiments?

Author Response: This is a great question. Previous ChIP-qPCR data (PMID: 20864036) indeed suggests that MSK1 can bind to the BMP2 promoter. To measure MSK1 occupancy at each locus (*PRKCB*, *BMP2*, *SHROOM2*, *ZNF462* and *GDF6*), we performed ChIP-qPCR at the promoter of each gene in WT and MSK1 KO HEK293T cells (using primers shown in **Figure S4**). Our experiments demonstrate that MSK1 is significantly ($P < 0.05$) enriched at the promoter regions of *BMP2* and *GDF6*, but not at the promoter regions of *PRKCB*, *SHROOM2*, or *ZNF462* (**Supplementary Figure 5**). This data, combined with our results that *BMP2* and *GDF6* can be activated by dCas9-dMSK1 and *PRKCB*, *SHROOM2*, and *ZNF462* cannot be activated by dCas9-dMSK1, collectively support our hypothesis that *BMP2* and *GDF6* are direct targets of MSK1, whereas *PRKCB*, *SHROOM2*, and *ZNF462* appear to be indirectly affected by the MSK1 KO (**Figure 2**). Nevertheless, we cannot exclude the possibility that MSK1 may be binding elsewhere within the promoter regions of *PRKCB*, *SHROOM2*, and *ZNF462*. We have carefully remodified our language in the revised manuscript to reflect these new data and possibilities.

Page 9, line 194: The authors validate the induction of target genes in response to MSK1 recruitment. It would be important to show that overexpression of one or two targets indeed mediates resistance to the BRAF inhibitor.

Author Response: We agree and thank Reviewer #2 for this important suggestion. We induced the upregulation of *EPDR1* or *AFF2* (the top 2 hits in **Figure 6**) using dCas9-dMSK1 (and corresponding gRNAs) in A375 cells and then treated cells with either DMSO (control) or 3.5 μ M PLX4720 for 72 hours. We then used microscopy to monitor cell growth and morphology and we used MTT assays to determine cell viability (**Supplementary Figure 18**). Our results indeed show that dCas9-dMSK1 mediated upregulation of *EPDR1* or *AFF2* results in improved cell fitness and higher viability compared to dCas9 control cells when challenged with the BRAF inhibitor PLX4720.

Page 4, line 73: typo

Author Response: We appreciate Reviewer #2 noting our oversight here and we have corrected this typo in our revised manuscript.

Reviewer #1 (Remarks to the Author):

The authors have done a great deal of work to improve the quality of this article. However, I am disappointed that they excluded two key studies that speak to the relationship between H3S28ph and H3K27ac (PMID: 27768872, 28077620). The authors' results confirm the findings of these studies.

I find the contributions of this study to the literature as incremental. Sure the technique is clever, but what new have we learned about the mechanism of action, not much. We know that H3S10ph and H3S28ph recruit 14-3-3, a protein that is required in the action of these phosphorylated PTMs. There is nothing about these 14-3-3 proteins in this study.

Why does MSK select to phosphorylate S10 or S28? Nothing new is offered in this study but simply to repeat what has been shown many times in the literature that MSK does one or the other and rarely both.

MSK recruitment to an upstream promoter region is thought to establish a permissive environment for chromatin remodeling, protein modification and transcription factor binding. The authors' findings add little to what we already know.

Reviewer #2 (Remarks to the Author):

The authors have satisfactorily addressed all my concerns.

AUTHORS' RESPONSE TO REVIEWER COMMENTS

Reviewer #1 (Remarks to the Author):

The authors have done a great deal of work to improve the quality of this article. However, I am disappointed that they excluded two key studies that speak to the relationship between H3S28ph and H3K27ac (PMID: 27768872, 28077620). The authors' results confirm the findings of these studies.

I found the contributions of this study to the literature as incremental. Sure the technique is clever, but what new have we learned about the mechanism of action, not much. We know that H3S10ph and H3S28ph recruit 14-3-3, a protein that is required in the action of these phosphorylated PTMs. There is nothing about these 14-3-3 proteins in this study.

Why does MSK select to phosphorylate S10 or S28? Nothing new is offered in this study but simply to repeat what has been shown many times in the literature that MSK does one or the other and rarely both.

MSK recruitment to an upstream promoter region is thought to establish a permissive environment for chromatin remodeling, protein modification and transcription factor binding. The authors' findings add little to what we already know.

Author Response

*We appreciate this input from Reviewer #1 and thank the reviewer for their time, suggestions, and expertise, all of which have greatly improved this manuscript. We emphasize that our work here is more centered upon the dCas9-dMSK1 CRISPR-based technology and the optimal methods for use thereof. Although we cited both PMID 27768872 and PMID 28077620 in our previous resubmission, we have revised our manuscript further to include and cite these key publications. Specifically, within the newly modified sentence in line 178: "H3S28ph has previously been shown to influence, and be influenced by, the dynamics of surrounding histone PTMs, especially the acetylation status of histone H3 lysine 27 (H3K27ac)."

*We thank Review #1 for their insights about the role of 14-3-3 proteins. We recognize that several studies show that 14-3-3 proteins bind to phosphorylated H3S10 and H3S28 and in so doing can facilitate transcriptional activation (e.g. PMID: 18418070, 18059471, and 21524388). Although we have not performed any experiments to verify the role of 14-3-3 proteins in dCas9-dMSK1 mediated histone phosphorylation and gene activation, we anticipate that future studies aimed at identifying the auxiliary factors supporting dCas9-dMSK1 activity will be crucial for resolving this important issue. Such studies are beyond the scope of our current work.

*Presently we do not know why natural full-length MSK1 appears to selectively phosphorylate either H3S10 or H3S28. Previous studies indicate that H3S10ph and H3S28ph do not coexist at the promoters of immediate early genes (PMIDs 20129940 and 21282660). However, it has also been reported that H3S10ph and H3S28ph can coexist at the α -globin promoter (PMID 21282660). It is possible that MSK1 can selectively write either mark or deposit both marks depending upon the chromatin or locus context and/or upon the presence of chromatin associated factors. Regardless, our data indicate that H3S10ph and H3S28ph do not coexist at the promoters of *OCT4* or *MYOD* when targeted by dCas9-dMSK1, which agrees with a large body of prior work.

*We agree with Reviewer #1 that MSK recruitment to an upstream promoter region can act to establish a permissive environment for chromatin remodeling, protein modification, and transcription factor binding. In fact, these properties were strong motivators for our selection of MSK1 as a component of this programmable histone kinase tool. Interestingly, we observe that full-length MSK1 does not harbor robust transactivation potential when fused to dCas9, whereas the truncated version of MSK1 (dMSK1) does. dCas9-dMSK1 is the first CRISPR-based histone H3 kinase, and as such holds tremendous potential for investigating the important mechanisms and functions of endogenous human histone phosphorylation, such as how full-length MSK1 and dMSK1 might differentially remodel targeted chromatin. While important and exciting, these studies are beyond the scope of the current manuscript.

Reviewer #2 (Remarks to the Author):

The authors have satisfactorily addressed all my concerns.

Author Response

We thank Reviewer #2 for their time and helpful input in reviewing this manuscript. We believe that the suggestions from Reviewer #2 substantially improved our manuscript and strengthened our findings.